# Electrochemical Detection of H_2_O_2_ Using Bi_2_O_3_/Bi_2_O_2_Se Nanocomposites

**DOI:** 10.3390/nano14191592

**Published:** 2024-10-02

**Authors:** Pooja D. Walimbe, Rajeev Kumar, Amit Kumar Shringi, Obed Keelson, Hazel Achieng Ouma, Fei Yan

**Affiliations:** Department of Chemistry and Biochemistry, North Carolina Central University, Durham, NC 27707, USA; pwalimbe@eagles.nccu.edu (P.D.W.); ashringi@nccu.edu (A.K.S.); okeelson@eagles.nccu.edu (O.K.); houma@eagles.nccu.edu (H.A.O.)

**Keywords:** electrochemical sensors, nanomaterials, hydrogen peroxide, bismuth oxyselenide

## Abstract

The development of high-performance hydrogen peroxide (H_2_O_2_) sensors is critical for various applications, including environmental monitoring, industrial processes, and biomedical diagnostics. This study explores the development of efficient and selective H_2_O_2_ sensors based on bismuth oxide/bismuth oxyselenide (Bi_2_O_3_/Bi_2_O_2_Se) nanocomposites. The Bi_2_O_3_/Bi_2_O_2_Se nanocomposites were synthesized using a simple solution-processing method at room temperature, resulting in a unique heterostructure with remarkable electrochemical characteristics for H_2_O_2_ detection. Characterization techniques, including powder X-ray diffraction (XRD), X-ray photoelectron spectroscopy (XPS), and scanning electron microscopy (SEM), confirmed the successful formation of the nanocomposites and their structural integrity. The synthesis time was varied to obtain the composites with different Se contents. The end goal was to obtain phase pure Bi_2_O_2_Se. Electrochemical measurements revealed that the Bi_2_O_3_/Bi_2_O_2_Se composite formed under optimal synthesis conditions displayed high sensitivity (75.7 µA µM^−1^ cm^−2^) and excellent selectivity towards H_2_O_2_ detection, along with a wide linear detection range (0–15 µM). The superior performance is attributed to the synergistic effect between Bi_2_O_3_ and Bi_2_O_2_Se, enhancing electron transfer and creating more active sites for H_2_O_2_ oxidation. These findings suggest that Bi_2_O_3_/Bi_2_O_2_Se nanocomposites hold great potential as advanced H_2_O_2_ sensors for practical applications.

## 1. Introduction

Electrochemical detection of hydrogen peroxide (H_2_O_2_) is a widely used technique in various fields, including analytical chemistry, biochemistry, and environmental monitoring. It relies on the principle of producing a measurable electrical signal due to the oxidation and reduction of H_2_O_2_ at the surface of the electrode. This method is appealing and has gained attention in recent studies over the other traditional techniques like chromatography [1], colorimetry [2], fluorimetry [3], etc. due to its remarkable advantages such as high sensitivity, compatibility, low cost, and real-time analysis. The current standard methods for H_2_O_2_ detection, such as colorimetric assays, UV–Vis spectrophotometry, and fluorescence-based techniques, face several significant limitations that hinder their performance. One major challenge is their limited sensitivity and low detection limits, making it difficult to accurately measure H_2_O_2_ at very low concentrations, particularly in the sub-micromolar range required for biological or environmental monitoring. These methods often suffer from interference by other substances commonly found in biological samples, such as ascorbic acid, uric acid, and glucose, which can generate false positives or obscure the true H_2_O_2_ signal. Moreover, these techniques typically require complex sample preparation, which may involve additional reagents or enzyme reactions that introduce variability and reduce reproducibility. For example, enzymatic assays that rely on horseradish peroxidase (HRP) often exhibit instability due to enzyme degradation over time. Additionally, these traditional approaches are generally time-consuming, require expensive equipment, and are not easily adapted for real-time or in situ measurements, which limits their practical application in dynamic environments like living cells or industrial processes. Furthermore, their relatively poor selectivity can lead to cross-reactivity, especially in complex biological matrices, resulting in less reliable data. These constraints highlight the need for more advanced, sensitive, and selective methods for H_2_O_2_ detection.

Electrochemical detection is usually carried out using enzymatic and non-enzymatic methods. However, enzymatic electrochemical sensors are expensive, less stable due to enzyme denaturation, show poor reproducibility, and require a tedious purification process. Because of these drawbacks, the non-enzymatic H_2_O_2_ electrochemical sensors [4] are in demand, as they offer high reproducibility, selectivity, and sensitivity. Moreover, they are more stable, easy to use, and inexpensive. The electrochemical detection of hydrogen peroxide (H_2_O_2_) involves redox reactions where H_2_O_2_ either gets oxidized or reduced at the surface of an electrode, depending on the applied potential and the nature of the electrode material. In the oxidation process, H_2_O_2_ donates electrons and decomposes into oxygen and protons, while, in the reduction process, it accepts electrons, breaking down into water. The electrode material plays a crucial role in facilitating these reactions: noble metals like platinum (Pt) and gold (Au) are often used due to their excellent catalytic properties, while carbon-based electrodes and enzyme-modified surfaces (such as those with horseradish peroxidase) are employed to enhance sensitivity and selectivity. These materials help convert the chemical signal (the interaction of H_2_O_2_ with the electrode) into an electrical signal, which can then be measured to quantify the concentration of H_2_O_2_

The electroreduction potential of hydrogen peroxide (H₂O₂) [5,6] varies depending on factors such as the electrode material, pH of the solution, and other experimental conditions [7]. However, in general, the reduction of H_2_O_2_ occurs at relatively low potentials. In acidic conditions (pH < 7), the typical electroreduction reaction is $H_{2}O_{2}+2H^{+}+2e^{-}\to 2H_{2}O$. The reduction potential is usually around +0.3 V to +0.6 V vs. the standard hydrogen electrode (SHE). In neutral or basic conditions (pH > 7), the reduction reaction changes to $H_{2}O_{2}+2e^{-}\to 2OH^{-}$. The potential shifts to more negative values, typically ranging from 0 V to −0.1 V vs. SHE. The exact potential depends on the electrode material used, with some electrodes, such as platinum or gold, showing electrocatalytic activity that can reduce the overpotential needed for H_2_O_2_ reduction.

Nowadays, metal oxide nanoparticles are used for H_2_O_2_ sensors [8], which offer distinctive catalytic and electrical properties and excellent stability for the redox reaction of hydrogen peroxide. In this respect, bismuth-based materials like bismuth oxide [9,10], bismuth chalcogenide [11,12,13,14], bismuth oxychloride [15], etc. have gained significant interest in the fields of energy storage, photocatalysis, and electrochemical sensing applications. Bismuth-based materials are increasingly popular in sensing technologies due to their unique combination of physical, chemical, and electrical properties. Their high sensitivity stems from their excellent electrical conductivity, particularly in compounds like bismuth oxide (Bi_2_O_3_), which enables rapid response and high precision in detecting analytes such as gases, biomolecules, and heavy metals. Bismuth-based sensors are widely used in gas sensing [16,17] for pollutants like nitrogen dioxide (NO_2_) and carbon dioxide (CO_2_) because of their high surface area, which enhances gas adsorption and sensitivity. One of the key advantages of bismuth-based materials is their eco-friendliness. Bismuth is non-toxic compared to other heavy metals like lead, mercury, or cadmium, making it a safer [18,19] and more environmentally benign option for sensing applications. This is especially important in biomedical sensors, where materials interact with biological tissues, as well as in environmental sensors that monitor pollutants without contributing additional hazards.

Bismuth compounds also show good stability under varying environmental conditions, such as temperature and humidity, which makes them robust for long-term sensing applications. Moreover, their electrochemical properties are highly advantageous for detecting heavy metals in water, as they can efficiently catalyze reactions at low potentials, providing high sensitivity and selectivity for metal ions like lead (Pb²⁺) or cadmium (Cd²⁺). Another important advantage of bismuth-based materials is their versatility. These materials can be synthesized in various nanostructured forms, including nanoparticles, nanowires, and thin films, which further enhance their surface area and reactivity, making them adaptable for diverse sensing platforms. This structural flexibility allows for fine-tuning of their properties, enabling customized sensor designs for specific applications in environmental monitoring, healthcare diagnostics, and industrial process control.

An emerging bismuth-containing nanomaterial that shows distinguished thermal, chemical and optoelectronic properties is bismuth oxyselenide (Bi_2_O_2_Se). Bi_2_O_2_Se is a two-dimensional layered material where [Bi_2_O_2_]^2+^ and Se^2−^ ions are held together by weak electrostatic forces [20,21]. To date, various synthesis methods have been developed to produce Bi_2_O_2_Se nanosheets, and among these, chemical vapor deposition [22], pulsed laser deposition [23], one-pot hydrothermal method [24], and solution-phase method [25] are the most popular. In most cases, selenourea is utilized as the Se source. In this work, we synthesized Bi_2_O_2_Se using an approach developed by Chitara et al. [26,27] with slight modifications. Here, as a selenium source, we used selenium powder in place of selenourea. This modified protocol provided an easy, scalable, and cost-effective synthetic recipe to obtain Bi_2_O_2_Se nanosheets. The use of selenium powder (USD 13/g) instead of selenourea (USD 93/g) significantly reduces the cost of synthesis. Moreover, the toxicity of selenourea is also significantly higher. Our work also identifies the optimal conditions to grow pure phase Bi_2_O_2_Se nanosheets from Se powder under mild solution-processing conditions. Furthermore, we explore the efficacy of the intermediate Bi_2_O_3_/Bi_2_O_2_Se (or Bi_2_O_x_Se_y_) nanocomposites (prepared by varying the synthesis time from 10 min to 7 days) towards H_2_O_2_ sensing. Unlike the conventional catalytic sensing, herein, we observe that the reduction peaks close to −0.7 V are influenced by the amount of H_2_O_2_ present in the electrolyte. The change in the current response is an ‘inhibitive type’. A possible mechanism is proposed towards this interaction of Bi_2_O_x_Se_y_ with H_2_O_2_.

## 2. Material and Methods

### 2.1. Materials and Reagents

Bismuth nitrate pentahydrate (Bi(NO_3_)_3_·5H_2_O) (Thermo Scientific, Fair Lawn, NJ, USA), hydrazine hydrate (N_2_H_4_·H_2_O, Hydrazine, 64%) (Thermo Scientific, Fair Lawn, NJ, USA), selenium powder (Alfa Aesar), potassium hydroxide (Fischer Scientific, Pittsburgh, PA, USA), sodium hydroxide (Fischer Scientific, Pittsburgh, PA, USA), ethylenediaminetetraacetic acid disodium salt (EDTA, Across Organics, Waltham, MA, USA), Nafion solution (5% in lower alcohols, Sigma Aldrich, St. Louis, MO, USA), isopropyl alcohol (Merck St. Louis, MO, USA), and deionized water (DI water, <18 Ω) were used for the experiments. H_2_O_2_ (30%, Fischer Scientific, Pittsburgh, PA, USA) was used as the analyte. Phosphate-buffered saline (PBS, pH 7.4, 1X molarity) was used as an electrolyte medium.

### 2.2. Synthesis

The Bi_2_O_x_Se_y_ nanomaterials were synthesized using a solution-phase method at room temperature, incorporating some modifications from an earlier work by Chitara et al. [26,27]. First, 1 g Bi(NO_3_)_3_·5H_2_O was dissolved in 200 mL of deionized water and sonicated for 30 min. Then, another solution containing 80 mg of selenium powder with 10 mL hydrazine hydrate was added into the bismuth nitrate solution. The solution turns black. In this mixture, 3 g of EDTA was added, and the stirring was continued for another 10 min. Finally, 1.2 g of KOH and 3.2 g of NaOH were added. The solution was vigorously stirred for a certain reaction time. The obtained precipitate was then centrifuged at 5000 rpm, washed with water and ethanol, and dried in an oven at 80 °C. To obtain variable compositions of the Bi_2_O_3_/Bi_2_O_2_Se (Bi_2_O_x_Se_y_) nanocomposites, the final reaction time (post-addition of KOH and NaOH) was varied, viz., 10 min, 3 h, 6 h, 18 h, and 7 days. The samples are denoted as BOSe-10 min, BOSe-3 h, BOSe-6 h, BOSe-18 h, and BOSe-7 days, respectively. The instrument details for characterization are provided in the Appendix A (ESI).

### 2.3. Modification of GCE and Electrochemical Sensing

For H_2_O_2_ sensing, we used a 3-electrode configuration with glassy carbon electrode (GCE, 3 mm diameter, 0.07 cm^2^), platinum wire (Pt) as the counter electrode, and Ag/AgCl (3.5 M KCl) as the reference electrode. The voltametric cell was placed in a cell stand (C3, BAS Inc., West Lafayette, IN, USA) equipped with controlled stirring and argon gas purging. Cyclic voltammetry (CV) and differential pulse voltammetry (DPV) signals were recorded to examine the sensing performance towards H_2_O_2_ and other analytes (uric acid, NaCl, ascorbic acid (AA), and dopamine).

For electrochemical analyses, dispersed solutions of the nanocomposites were prepared by mixing 5 mg of the samples in isopropanol/DI water (9:1 ratio), along with 100 μL of Nafion solution as the binder. This dispersion (2.5 μL) was then drop cast on polished GCE. For the electrochemical measurements, 15 mL of phosphate-buffered saline (PBS 1X, pH 7.4) was taken in the voltametric cell and purged with argon gas for 30 min, with stirring at 400 rpm. During the electrochemical measurements, the stirring and purging was momentarily paused to record the CV and DPV scans. The potential window was 0.8 to −0.9 V vs. Ag/AgCl. The scan rate used was 10 mV/s. The CV scan starts from 0 V, and is scanned up to 0.8 V in the positive direction, followed by the scan in the negative direction up to −0.9 V. The cycle is completed by scanning back to 0.8 V. The pulse parameters for DPV, i.e., height, width, period, and increment, were 50 mV, 0.01 s, 0.1 s, and 10 mV, respectively. The DPV scan direction is negative, i.e., is from 0.8 to −0.9 V. The DPV scans were recorded without applying any inversion during the cathodic and anodic sweeps. The initial CV and DPV scans of the electrode, in the blank electrolyte, i.e., without H_2_O_2_ addition, were performed prior to the sensing tests. Then, 150 µL of H_2_O_2_ from 2 µM stock solution was added. The effective concentration of H_2_O_2_ in the electrolyte was 100 times lower. Purging and stirring were continued for 5 min for each addition of H_2_O_2_. The stock solutions of 2, 20, 200, 1000, and 10,000 µM were used for H_2_O_2_ sensing. The effective concentration at each addition of H_2_O_2_ was calculated and used in the sensitivity (S) evaluation.

For the selectivity tests, analytes of 10 mM concentrations (stock) were used, except for uric acid (0.4 mM stock, since it is the maximum dissolvability in water). First, 150 µL of these stock solutions were sequentially added. The concentration of the interferant analytes (uric acid, NaCl, ascorbic acid (AA), and dopamine) was purposefully kept high to study the efficacy of the sample towards H_2_O_2_ detection. In biological fluids like sweat and urine, various analytes are commonly present. For a non-invasive analysis of H_2_O_2_, which indirectly estimates glucose, electrochemical measurements in the presence of these interferants offer a more realistic comparison to real samples. Interferants can sometimes interact with the sensor material, generating additional peaks in CV curves. While an ideal sensor would be highly selective for a single analyte, this is often not the case in practice. Uric acid, NaCl, ascorbic acid, and dopamine are considered interferents in H_2_O_2_ sensing, because they can generate electrochemical signals similar to or overlapping with H_2_O_2_, thus reducing the sensor specificity and accuracy. Uric acid and ascorbic acid are both electrochemically active and possess redox potentials close to that of H_2_O_2_, meaning they can be oxidized or reduced at similar potentials, leading to false-positive signals or noise in the detection system. Dopamine, another electroactive compound, undergoes redox reactions (close to 0 V vs. Ag/AgCl) that can interfere with the H_2_O_2_ signal, particularly in biosensors or electrochemical sensors. NaCl, though not electroactive, can affect the ionic strength and conductivity of the solution, altering the sensor’s baseline response and making it harder to distinguish H_2_O_2_ signals. These compounds are often found in biological fluids, such as blood or urine, where H_2_O_2_ is being detected, making it critical to minimize or account for their interference to ensure reliable H_2_O_2_ measurements.

## 3. Results and Discussion

### 3.1. Structural and Morphological Characterization

The powder XRD patterns are shown in Figure 1. The standard data from the Inorganic Crystal Structure Database (ICSD) are also provided for reference. The sample prepared with a reaction time of 10 min showed the formation of α-Bi_2_O_3_ (ICSD code: 2374), i.e., monoclinic structure with space group P 121/c1. With increasing the reaction time, the formation of the Bi_2_O_2_Se-type phase is observed. The intensity of the Bi_2_O_2_Se component increases with a longer reaction time, and finally, at 7 days, it completely transforms into a single phase. The crystal structure of the Bi_2_O_2_Se is tetragonal (I 4/mmm space group, ICSD code: 2903), similar to that reported by Ghosh et al. [25]. For simplicity, we refer to the intermediate nanocomposite with varying Se content as Bi_2_O_x_Se_y_. Appendix A (ESI) shows the XRD patterns with higher amounts of Se precursor keeping the synthesis time of 18 h.

The normalized UV–Vis-NIR diffuse reflectance spectra (UV–Vis-NIR DRS) of the samples are shown in Figure 1b. The BOSe-7 days sample shows a slightly different reflectance pattern (indicating a different phase) than the others. The composite samples show higher absorption in the NIR region but lower absorption in the UV–Visible region. Nevertheless, the band edge is slightly different, leading to similar direct but lower indirect band gaps, which are estimated using the Kubelka–Munk function [28,29]. Interestingly, from the Tauc plots, we can observe a slightly higher area below the linear region (Figure 1c,e), for BOSe-6h, which indicates a slightly higher proportion of defects in this sample compared to BOSe-7 days. This is understandable, considering more interfaces for the nanocomposite. The lower indirect band gap suggests better charge–transfer characteristics can be present in this sample.

The FE-SEM images at two different scales are shown in Figure 2. The BOSe-10 min sample has predominant micron-sized rod-like structures with very fine agglomerates on the surface. The lack of a sheet structure indicates the absence of any Bi_2_O_2_Se. With an increased reaction time (BOSe-3 and 6 h), the sheets appear to evolve from the surface of these rod-like structures. At around 18 h, the rods completely transform into agglomerated sheet-like structures. The sheets become well defined for BOSe-7 days, indicating complete transformation to Bi_2_O_2_Se. The elemental mapping and EDX analyses for the representative BOSe-6 h and BOSe-7 days are provided in Appendix A in the ESI. Clearly, the elemental analyses for the BOSe-7 days sample show a good match with the expected stoichiometry of Bi_2_O_2_Se. The BOSe-6 h sample has a much lower Se content, in line with the inference from XRD. The XPS analysis for this sample is also provided as Appendix A in the ESI, which could possibly be more accurate than EDX.

### 3.2. Growth Mechanism

Based on the evidence from XRD, SEM, EDX, and XPS, we can hypothesize the growth mechanism of the Bi_2_O_x_Se_y_ samples with this synthesis protocol. Equations (1)–(4), shown below, indicate the possible chemical reactions involved in the formation of Bi_2_O_2_Se nanosheets, similar to that suggested by Chitara et al. [26]. According to their mechanism, Bi(NO_3_)_3_ can undergo hydrolysis to produce BiONO_3_. Furthermore, selenium powder is reduced by hydrazine to produce Se^2−^ ions in a highly alkaline medium. Finally, the self-assembly of the oppositely charged (Bi_2_O_2_)^2+^ and Se^2−^ layers results in the formation of Bi_2_O_2_Se nanosheets.
(1)$$Bi{\left(NO_{3}\right)}_{3}+H_{2}O\to BiONO_{3}+2H^{+}+2NO_{3}^{-}$$
(2)$$2BiONO_{3}+OH^{-}\to {Bi}_{2}O_{3}+H^{+}+2NO_{3}^{-}$$
(3)$$2Se+N_{2}H_{4}+4OH^{-}\to 2{Se}^{2-}+N_{2}+4H_{2}O$$

Based on our evidence in this work, it appears that this mechanism may not be complete. We suggest that the availability of Se^2−^ ions could be a rate-limiting factor and the replacement of the Se^2-^ ions may be much slower, unlike previously anticipated. Therefore, we can expect the following two outcomes:


I.Large availability of Se^2−^

(4)
$$2BiONO_{3}+{Se}^{2-}\to {Bi}_{2}O_{2}Se+2NO_{3}^{-}$$




II.Low availability of Se^2−^

(5)
$$4BiONO_{3}+OH^{-}+{Se}^{2-}\to {Bi}_{2}O_{2}Se+{Bi}_{2}O_{3}+H^{+}+4NO_{3}^{-}$$



Another possible route could be through the formation of hydroxide or oxyhydroxide, as below:(6)$$2Bi{\left(NO_{3}\right)}_{3}+3H_{2}O\to {Bi}_{2}O_{3}+6H^{+}+6NO_{3}^{-}$$
(7)$${Bi}_{2}O_{3}+H_{2}O\to 2Bi{\left(OH\right)}_{3}$$
(8)$${Bi}_{2}O_{3}+12{OH}^{-}\to 2Bi{\left(OOH\right)}_{3}+3H_{2}O$$
(9)$$2Bi{\left(OH\right)}_{3}+{Se}^{2-}\to {Bi}_{2}O_{2}Se+6H^{+}+2O_{2}$$
(10)$$2Bi{\left(OOH\right)}_{3}+{Se}^{2-}\to {Bi}_{2}O_{2}Se+6H^{+}+5O_{2}$$

However, more work needs to be done to prove the mechanism, which will be undertaken in the future.

### 3.3. Electrochemical Sensing Performance

The CV and DPV scans were performed to analyze the sensitivity by applying a potential from 0.6 to −0.9 V against the Ag/AgCl electrode. First, 150 µL of hydrogen peroxide solution with different concentrations (ranging from 2–10,000 µM stock) were then added into 15 mL of argon-purged PBS (pH 7.4, 1X). The effective concentrations were calculated to range from 0.02 to 410 µM. In between each addition, stirring and purging were employed to help mix and maintain the inert atmosphere inside the cell. For each composite, before the addition of H_2_O_2_, blank reading, i.e., without H_2_O_2_, was performed. Appendix A (ESI) depicts the cyclic voltammograms of the different samples. For visual clarity, only selective scans are plotted in Figure 3andAppendix A. The change in the redox peak patterns is clearly visible among the samples and also with the increasing H_2_O_2_ additions. However, we refrain from an in-depth evaluation of sensitivity (S) using CV, as the peak patterns are complex with multiple peaks.

The DPV technique is more sensitive than CV [30], as it excludes the capacitive contribution. The changes in the current values become significant even at low concentrations. Moreover, in this work (Figure 3), the peak profile is highly simplified (a single peak in the potential window of 0 to −0.9 V), unlike CV. Therefore, for the evaluation of sensitivity (S), we consider the DPV scans with this peak centered around −0.75 V. ‘S’ can be obtained from the plot of the current peak with respect to the concentration of H_2_O_2_ [30,31], as per the equation $S=\frac{k}{A}$ , where k is the slope of the linear fits, and A is the area of the working electrode in cm^2^. The calibration plot for the representative BOSe-6 h sample is provided in Figure 3f. The calibration curves for the other samples are provided in Appendix A (ESI). The plot in Figure 3f is obtained after considering the average of three independent experiments (Appendix A (ESI)) with fresh electrode–electrolyte setups. 

The evaluated sensitivity values are given in Table 1. The BOSe-6 h sample shows the highest sensitivity of 75.7 µA µM^−1^ cm^−2^ within the linear concentration range of 0–15 µM. It is interesting to note that the sensitivity values are significantly less for the sample prepared in 10 min (final stirring post addition of KOH and NaOH) and pure phase Bi_2_O_2_Se. These results clearly indicate that the efficacy of the composite Bi_2_O_x_Se_y_ (in particular, with a reaction time of 6 h) is much higher as compared to the pure phase Bi_2_O_2_Se towards H_2_O_2_ sensing. An additional experiment was performed without any purging using BOSe-6h sample and the results are provided in Appendix A (ESI). To ascertain the oxygen reduction activity, linear sweep voltammograms were also recorded using a rotating ring disk electrode (Appendix A (ESI)).

To ascertain the reusability of the electrode, we carried out three independent experiments with the same electrode (BOSe-6 h) but with fresh electrolytes (Figure 4). The electrode was mildly flushed with DI water and dried before each experiment. The electrode depicts results similar to those in Figure 3c. The CV scans also reveal similar characteristics each time. The decreasing trend in the current response is observed in all measurements with the increasing H_2_O_2_ amounts. A comparison, regarding some performance parameters for H_2_O_2_ sensing, with some recent reports is provided in Table 2.

### 3.4. Possible Mechanism of the Electrochemical Interaction with H_2_O_2_

We hypothesize that these redox peaks are from the metal oxide and metal selenide. We observe that, for all samples, there is an absence of the reduction peak in the first CV cycle. Oxidation peaks appear in the first cycle itself. In the subsequent cycle, prominent reduction peak is also observed. These reduction and oxidation peaks are intensified upon H_2_O_2_ addition. This suggests that some sort of chemical reactions occur between the electrolyte and electrode material. The redox peaks most probably belong to the metal oxide/selenide composite and are not necessarily from the H_2_O_2_ electroreduction. In the presence of H_2_O_2_, the selenide component can become preferentially oxidized and lead to a more intense peak for the oxide component. The shift in DPV peak position can also occur due to changes in the electrode material during each cycle.

### 3.5. Electrochemical Response in the Presence of Different Analytes

For the selectivity test, we utilize the BOSe-6 h as a representative sample. CV and DPV scans were recorded with the sequential addition of high concentrations of interfering analytes (effective concentration is 100 µM for all, except uric acid, which is 2 µM) in the same electrolyte. As seen in Figure 5, the addition of uric acid, NaCl, and Ascorbic acid shows some minor change in the current signal. However, we observe more significant change when dopamine is added. The DPV peak response for dopamine is usually around 0 V (a small signal is observed; Figure 5b). The change is more significant with the H_2_O_2_ addition (around −0.7 to −0.8 V), indicating a higher selectivity for H_2_O_2_ over dopamine. To gain a better idea, we carried out the DPV analyses using these interferants individually. The sample is relatively less responsive to uric acid, NaCl, and ascorbic acid as compared to H_2_O_2_ for high concentrations. The minor changes can be attributed to the structural changes on the electrode surface with each cycle. However, the addition of dopamine is more interesting. An additional DPV peak appears close to 0 V, and the current response at −0.7 to −0.8 V actually increases with high concentrations (comparing it to H_2_O_2_, we see an inhibitive response and lowering of the current signal). To ascertain the practicality in real biological samples, experiments were performed on urine samples of two individuals. The results are provided in Appendix A (ESI). 

H_2_O_2_ sensors operating in the 0–15 µM range have several important applications, particularly in biological, environmental, and industrial fields where precise, low-level detection is critical. In biomedical research for oxidative stress monitoring, H_2_O_2_ is a byproduct of cellular metabolic processes and plays a role in oxidative stress, which is linked to various diseases such as cancer, diabetes, and neurodegenerative conditions. In biological systems, the H_2_O_2_ levels are typically in the low micromolar range (0.1–10 µM) under normal physiological conditions. During oxidative stress, the concentrations can increase but often remain below 15 µM in localized cellular environments. Sensors in the 0–15 µM range are essential for monitoring H_2_O_2_ at the physiological level in biological systems, enabling researchers to study the roles of oxidative stress and redox biology in real time.

In biological assays, enzymes like glucose oxidase generate H_2_O_2_ as a product. Sensors with high sensitivity can detect low concentrations of H_2_O_2_, allowing for the accurate monitoring of enzymatic reactions in glucose sensing, lactate monitoring, or cholesterol detection. Enzymes such as glucose oxidase produce H_2_O_2_ in micromolar concentrations. For example, in glucose biosensors, H_2_O_2_ concentrations typically fall between 1 and 10 µM during enzyme-catalyzed reactions, making this range critical for accurate enzymatic activity measurements.

H_2_O_2_ is used in water treatment processes, and low concentrations may remain in treated water. In treated water, residual H_2_O_2_ concentrations are generally low, often below 10 µM. Monitoring systems aim to detect trace amounts (in the 0–15 µM range) to ensure that the residual H_2_O_2_ is at safe levels for human consumption and environmental safety. In industries such as paper and textiles, where H_2_O_2_ is used as a bleaching agent, the concentrations used are much higher (in the millimolar range). However, during quality control, residual H_2_O_2_ concentrations are reduced to micromolar levels (typically below 10 µM) to prevent product degradation or ensure worker safety, making the 0–15 µM sensor range important for detecting trace amounts after treatment.

Furthermore, H_2_O_2_ is often used as a disinfectant in food processing and sterilizing packaging. After use, residual H_2_O_2_ should be reduced to very low levels, typically below 5–10 µM, to ensure food safety. Sensors that can detect within this micromolar range are crucial for ensuring compliance with safety standards.

Many pharmaceuticals can produce reactive oxygen species (ROS) like H_2_O_2_ as part of their mechanism of action. The concentrations of H_2_O_2_ monitored are typically in the low micromolar range (1–10 µM), making sensors in the 0–15 µM range suitable for these applications. Thus, sensors in the 0–15 µM range can help monitor oxidative changes during drug testing or therapeutic applications to ensure the efficacy and safety of drug formulations. In all these applications, high sensitivity to low H_2_O_2_ concentrations is crucial for precision, safety, and effectiveness, making sensors with a detection range of 0–15 µM highly valuable.

## 4. Conclusions

In conclusion, Bi_2_O_x_Se_y_ nanocomposites with varying Se contents were successfully synthesized using a solution-phase method by adjusting the reaction time, resulting in morphological transformations from α-Bi_2_O₃ rods to Bi_2_O_2_Se nanosheets. Structural and electrochemical characterization, particularly via CV and DPV techniques, revealed that the nanocomposite synthesized with an optimal reaction time of 6 h exhibited good sensitivity (75.7 µA µM⁻¹ cm⁻²) and selectivity for H_2_O_2_ detection, making it a promising candidate for non-enzymatic sensor applications. The observed redox behavior, characterized by oxidation peaks in the first CV cycle, followed by intensified reduction peaks in subsequent cycles, which were modulated upon H_2_O_2_ addition, suggests ongoing interactions between the electrolyte and electrode material. Notably, the redox peaks are likely associated with the metal oxide/selenide composite rather than directly from H_2_O_2_ electroreduction. The preferential oxidation of the selenide component in the presence of H_2_O_2_ may enhance the oxide component’s peak intensity. Although this hypothesis requires further validation, it underscores the complexity of the redox processes in these nanocomposites. Future studies utilizing advanced techniques will focus on unraveling the precise chemical mechanisms at play and the impact of material changes on sensor performance, as indicated by the shifting DPV peak positions and lowering of the current response during successive cycles. This work highlights the potential of Bi_2_O_x_Se_y_ nanocomposites as sensitive and selective H_2_O_2_ sensors while also identifying areas for deeper exploration.

## Figures and Tables

**Figure 1 nanomaterials-14-01592-f001:**
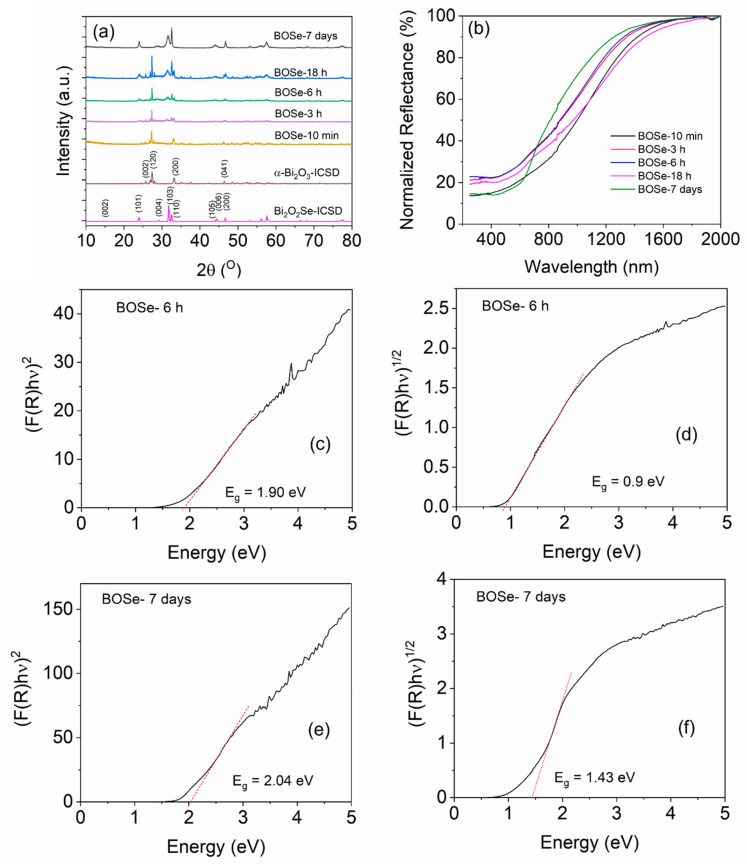
(**a**) XRD patterns for the synthesized samples. (**b**) UV–Vis-NIR DRS for the powder samples. Tauc plots for direct band gap estimation: (**c**) BOSe-6 h and (**e**) BOSe-7 days. Tauc plots for indirect band gap estimation: (**d**) BOSe-6 h and (**f**) BOSe-7 days.

**Figure 2 nanomaterials-14-01592-f002:**
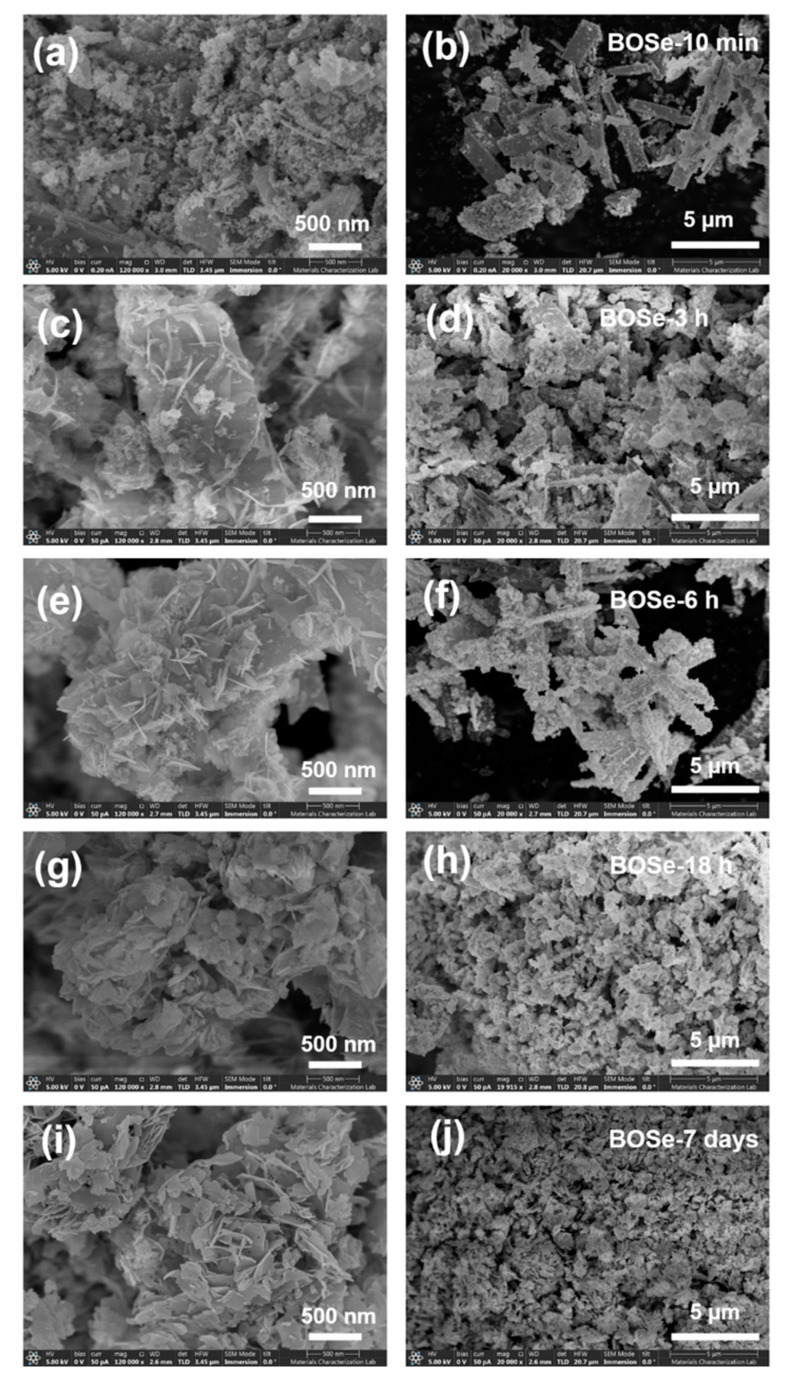
SEM images at two different scales, i.e., 500 nm (**left panel**) and 5 µm (**right panel**) for (**a**,**b**) BOSe-10 min, (**c**,**d**) BOSe-3 h, (**e**,**f**) BOSe-6 h, (**g**,**h**) BOSe-18 h, and (**i**,**j**) BOSe-7 days.

**Figure 3 nanomaterials-14-01592-f003:**
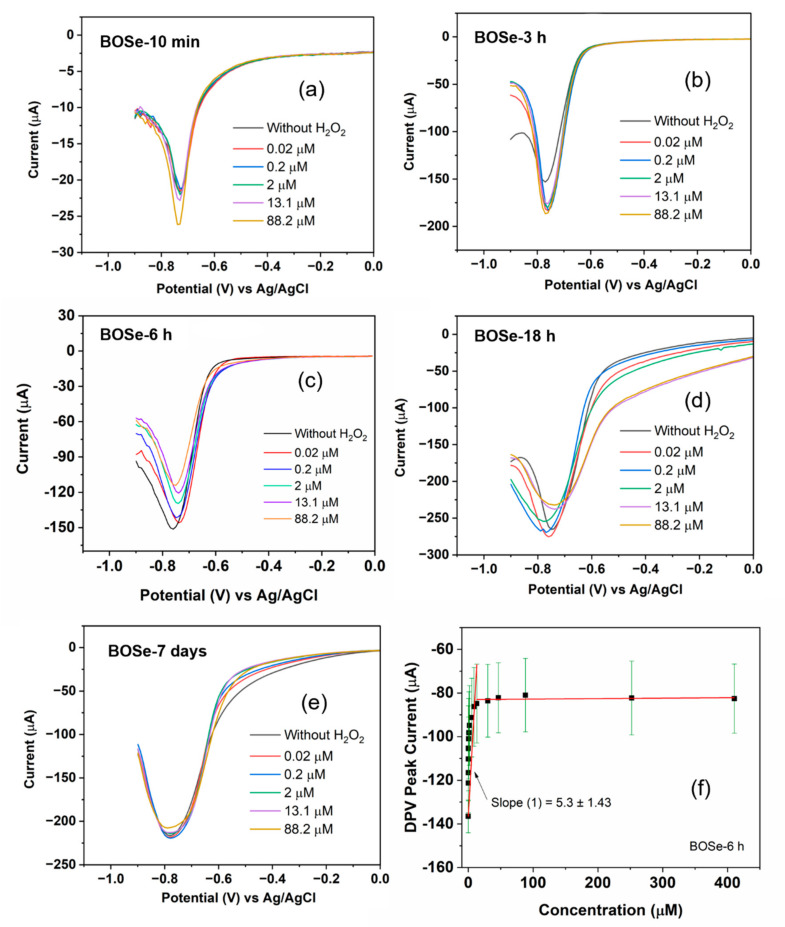
Selective DPV scans of Bi_2_O_x_Se_y_ samples with varying synthesis time of (**a**) 10 min, (**b**) 3 h, (**c**) 6 h, (**d**) 18 h, and (**e**) 7 days. (**f**) Calibration plot using the complete DPV data. The black squares are the averaged data for 3 independent experiments, and the red lines are the linear fits, while the standard errors are shown as green color lines.

**Figure 4 nanomaterials-14-01592-f004:**
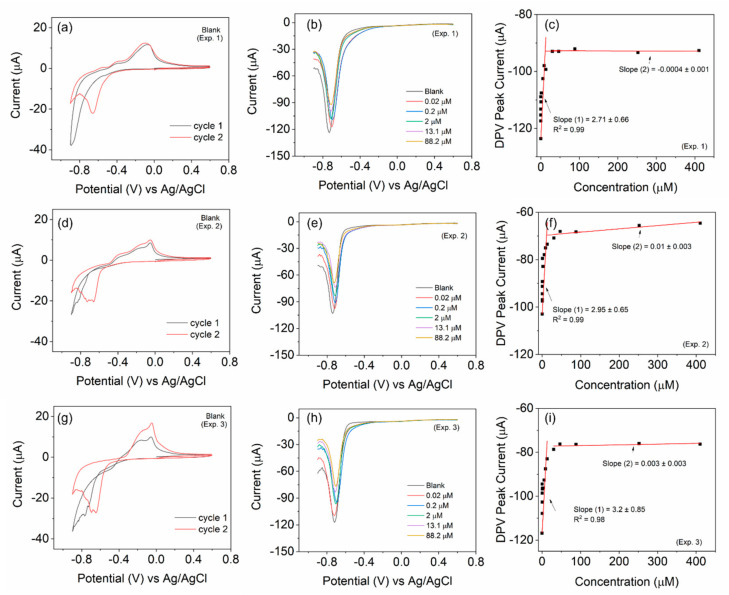
Reusability test of the BOSe-6 h sample carried out with the same electrode each time in fresh electrolyte. Three experiment sets for (**a**,**d**,**g**) CV scans at 10 mV/s for the blank (without H_2_O_2_), (**b**,**e**,**h**) selective DPV scans with varying H_2_O_2_ amounts, and (**c**,**f**,**i**) calibration plots with the complete data set of DPV.

**Figure 5 nanomaterials-14-01592-f005:**
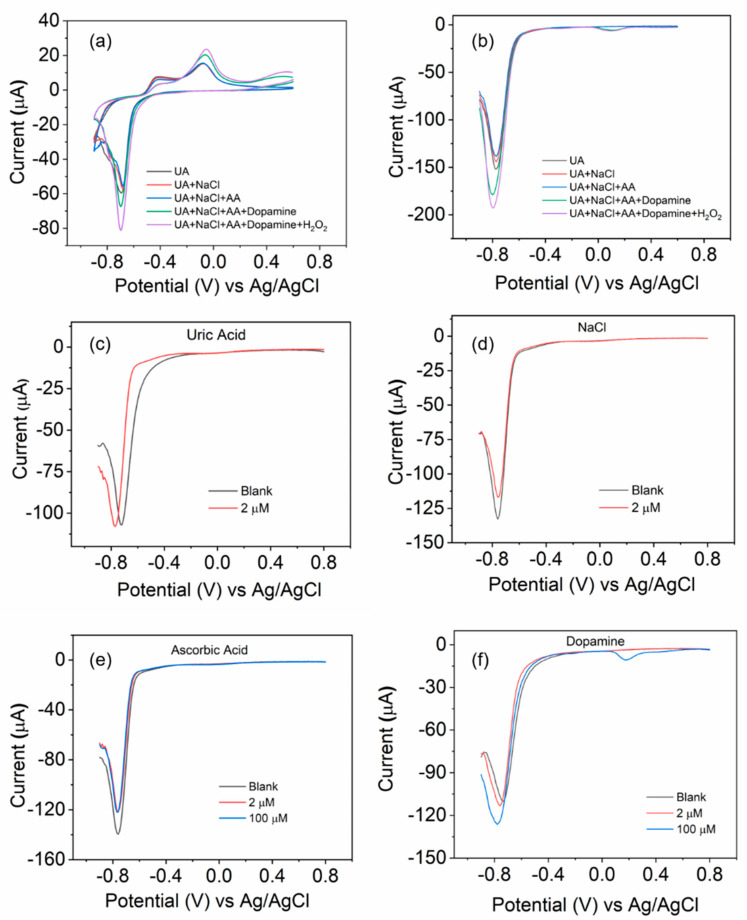
Electrochemical tests with the sequential addition of different interfering analytes using the BOSe-6 h sample: (**a**) CV scan at 10 mV/s and (**b**) DPV scans. The stock concentration of all analytes was 10 mM (except uric acid at 0.4 mM), of which 150 µL was added to 15 mL PBS. DPV scans of fresh samples in separate electrolyte solutions spiked with (**c**) uric acid (2 µM), (**d**) NaCl (2 µM), (**e**) ascorbic acid (2 and 100 µM), and (**f**) dopamine (2 and 100 µM).

**Table 1 nanomaterials-14-01592-t001:** Sensitivity values obtained from DPV scans.

Sample Code	Slope	Sensitivity (µA µM^−1^ cm^−2^)
BOSe-10 min	0.03	0.38
BOSe-3 h	0.37	5.29
BOSe-6 h	5.30	75.7
BOSe-18 h	3.04	43.4
BOSe-7 days	0.57	8.14

**Table 2 nanomaterials-14-01592-t002:** Comparison of the electrochemical sensing parameters with some published literature.

Sensing Material	Sensitivity	Detection Range	Limit of Detection (LoD)	pH	Reference
GC/Chi-(CXBiFe-1050)	4.55 μA mM^−1^	50–1000 µM	2.5 μM	7	[32]
GC/Chi-XGN	--	1–10 μM	0.36 μM	7	[33]
GC/Chi-CXBiFe_1.2_	--	3–30 μM	0.24 μM	7	[34]
Fe-doped carbon aerogel (Fe-CA)	1.78 mA M^−1^	1–50 mM	0.5 mM	7	[35]
Bi_2_O_3_/MnO_2_	0.914 µA µM^−1^cm^−2^	0.2–290 μM	0.05 μM	7.2	[36]
NF/HRP/Bi_2_O_3_–MWCNT/GCE	26.54 μA mM^−1^ cm^−2^	8.34–28.88 mM		7	[37]
Bi_2_Se_3_–Hb–Nf/GCE	--	2.0–100 μM	0.63 μM	7	[11]
Bi_2_S_3_/g-C_3_N_4_	1011 µA/ µM^−1^cm^2^	0.5–950 μM	0.078 μM	12	[12]
BOSe-6 h (with Ar purging)	75.7 µA µM^−1^ cm^−2^	0–15 μM	-	7.4	This Work
BOSe-6 h (no purging)	116 µA µM^−1^ cm^−2^	0–15 μM	-	7.4	This Work

## Data Availability

Data will be made available on request.

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
