# Peer review of "Electrochemical Detection of H_2_O_2_ Using Bi_2_O_3_/Bi_2_O_2_Se Nanocomposites"

_nanomaterials, 2024, doi:10.3390/nano14191592_

Round 1
Reviewer 1 Report
Comments and Suggestions for Authors
see attached

Author Response
The authors report synthesis of Bi2OxSey nanocomposites with varying Se content via a solution-phase method by varying the reaction time. The prepared nanostructure demonstrates great potential for H2O2 detection. I have the following comments to improve the manuscript:
We thank the reviewer for recommendation of our work and useful insights.
Major issues:
- Introduction: the authors should elaborate more on the advantages/benefits of using bismuth-based materials for sensing applications.
A: We have modified the introduction as suggested.
- Could the authors clarify the rationale behind choosing uric acid, NaCl, ascorbic acid, and dopamine as interferants for the selectivity tests?
A: In biological fluids like sweat and urine, various analytes are commonly present. For non-invasive analysis of H₂O₂, which indirectly estimates glucose, electrochemical measurements in the presence of these interferants offer a more realistic comparison to real samples. Interferants can sometimes interact with the sensor material, generating additional peaks in CV curves. While an ideal sensor would be highly selective for a single analyte, this is often not the case in practice. Therefore, it is essential to analyze the electrochemical response of a sensor material to multiple analytes.
- Reaction Mechanism: since the authors suggest that the availability of Se2- ion is the rate-limiting step, have the authors tested how varying the ratio of the starting materials affects the results?
A: In this work, for the desired composition Bi2O2Se, we need to keep the Bi: O: Se ratio as 2: 2: 1. A change in Se amount can lead to several intermediate compositions and possibly lead to formation of Bi2Se3, which is not our desired sample.
Nevertheless, as suggested by the reviewer we carried out additional synthesis using 240 mg (3X), 800 mg (10X), and 1600 mg (20X) in place of 80 mg of Se. The synthesis time was 18 h, keeping all other parameters same. The results are provided in the supporting information (Fig. S1). We observed the formation of amorphous Bi2O2Se, similar to Chitara et al. [1] and Ghosh et al. [2] for the sample prepared with 240 mg Se. This clearly proves that formation of Se2- ion is rate determining.
However, unreacted Se is clearly seen with higher amounts and there is complete absence of the desired Bi2O2Se phase. Thus, our initial hypothesis of ‘keeping mole ratio of starting precursors (as 2:2:1)’ is important and need not be tinkered.
- Chitara, B.; Limbu, T.B.; Orlando, J.D.; Vinodgopal, K.; Yan, F. 2-D Bi2O2Se Nanosheets for Nonenzymatic Electrochemical Detection of H2O2. IEEE Sensors Letters 2020, 4, doi:10.1109/LSENS.2020.3012300.
- Ghosh, T.; Samanta, M.; Vasdev, A.; Dolui, K.; Ghatak, J.; Das, T.; Sheet, G.; Biswas, K. Ultrathin Free-Standing Nanosheets of Bi2O2Se: Room Temperature Ferroelectricity in Self-Assembled Charged Layered Heterostructure. Nano Letters 2019, 19, 5703–5709, doi:10.1021/ACS.NANOLETT.9B02312
- Sensing Performance: How many replicates were conducted to determine the detection limit? Additionally, could the authors provide calibration curves for the samples with varying reaction time?
A: We had previously carried out one experiment for each sample. The results in Table 1 have been modified after careful re-fitting. Furthermore, in the revised manuscript we include the results for 3 data sets (using same electrode for repeatability) and 3 data sets (fresh electrodes for reproducibility), using a representative BOSe-6 h sample. The averaged calibration curve with the error bars for BOSe-6 h is provided in new Fig. 3 (f). The calibration curves for the other samples are provided as Fig. S6 in ESI. We believe that these additional data (for the best sample) should suffice the reviewer’s query.
- The authors tested sensor selectivity by sequentially adding high concentrations of interfering analytes. Have they considered performing these tests individually?
A: As suggested by the reviewer, we have carried out the selectivity tests individually for each analyte. The Fig. 5 is replaced with the new data for individual interferant along with sequential additional of these.
- What is the intended application of the H2O2 sensor, and what is the specific concentration of H2O2 required to sense for this application?
A: The H2O2 sensing with our material should be suitable for non-invasive biological samples like sweat and urine. The amount of H2O2 indirectly provides an estimate of the amount of glucose in a test subject. The usual concentration range of H2O2 in humans is 0.1-10 µM which is within the linear range provided by our samples.
For other applications, trace level detection of H2O2 can also be useful:
- Biomedical applications: 0.1–10 µM (oxidative stress and enzymatic reactions)
- Water quality monitoring: <10 µM (residual H₂O₂)
- Food safety: <5-10 µM (residual H₂O₂ in packaging)
- Pharmaceutical testing: 1-10 µM (ROS detection)
- Industrial quality control: <10 µM (residual H₂O₂ post-treatment)
Minor issues:
- Figure 5: The authors should provide explanation for subfigure (a) and (b) in the caption.
A: We have replaced Fig. 5 with a revised one. The details are provided in the caption.
Reviewer 2 Report
Comments and Suggestions for Authors
The manuscript “Electrochemical Detection of H2O2 using a BiO3/BiO2Se Nanocomposites” describe the fabrication and characterization of a Bi based sensor which is applied for the detection of H2O concentration in water-based solutions. The topic is interesting for the audience of the journal, but although the authors try to justify in the manuscript the novelty and the impact of the developed work, I consider that the novelty is not good enough to be published in this journal. I would highly recommend the authors to fully characterize and prove the mechanism occurring during the growth of the nanoparticles to fully understand and justify the behavior of the proposed sensor. Moreover, the range of interest for the hydrogen peroxide sensor in real applications is wider that the presented in this works, so the impact of the nanocomposite as a hydrogen peroxide sensor is also very limited, and it should be also improved.
Some comments that could be useful for the authors are:
1. In the introduction, the real interests of measuring the hydrogen peroxide and the range of interest should be longer discussed. The authors do not properly present the necessity to develop a sensor to detect hydrogen peroxide, and the real limitations of current standard methods.
2. In the introduction, the authors describe the disadvantages of using enzymatic methods to detect hydrogen peroxide and the advantages of using non-enzymatic methods. But they do not describe the mechanism of the hydrogen peroxide detection (potentials, why are interesting the use of bismuth, etc.). Some of the readers could not be familiarized with the electrochemical mechanism involved in the hydrogen peroxide detection. Please, you should clarify and extend this part too.
3. In the synthesis section, the authors do not clarify why they are using these chemicals, concentrations and solutions. It looks as a standard method or something from a previous work. Please, add a reference or explain it in the text.
4. Why are the authors using a glassy carbon electrode? What are the dimensions of the electrode? Why are thy using a purge solution to perform the electrochemical analysis? It could limit the real applicability in real scenarios of the sensor. What could happen if you do not purge the solution? The result and the applicability of the sensor could be reduced?
5. Why are you using those interferents to examinate the performance of the sensor? In which field are them interesting and in which range of concentrations? Clarify.
6. How reproducible is the dispersion to drop cast the solution in the electrode? Is this homogenous enough to guarantee the sensor performance? Do you need to consider some tricks or hidden steps to do it?
7. In the Structural and Morphological Characterization section there are some sentences in italics. The authors should correct it.
8. In the Figure 2, the scale and the conditions of the SEM analysis is too much small and impossible to read.
9. Sensitivity and limit of detection are not necessary are independent equations in the manuscript. They are very common, so the authors should include it as part of the text.
10. For BOSe-6h, why the peak of the hydrogen peroxide is shifting for increasing concentrations? Clarify in the discussion.
11. In general, the discussion of the results is quite poor and short, as well as the conclusions. More discussions and explanations about the results should be incorporated to the manuscript.
Comments on the Quality of English Language
Moderate editing of English language required.
Author Response
The manuscript “Electrochemical Detection of H2O2 using a Bi2O3/Bi2O2Se Nanocomposites” describe the fabrication and characterization of a Bi based sensor which is applied for the detection of H2O2 concentration in water-based solutions. The topic is interesting for the audience of the journal, but although the authors try to justify in the manuscript the novelty and the impact of the developed work, I consider that the novelty is not good enough to be published in this journal. I would highly recommend the authors to fully characterize and prove the mechanism occurring during the growth of the nanoparticles to fully understand and justify the behavior of the proposed sensor. Moreover, the range of interest for the hydrogen peroxide sensor in real applications is wider that the presented in this works, so the impact of the nanocomposite as a hydrogen peroxide sensor is also very limited, and it should be also improved.
We thank the reviewer for recommendation of our work and useful insights. This protocol that we have developed clearly suggests transformation of Bi2O3 to Bi2O2Se, using Se powder. Since, this is a first of a kind observation for this material using Se precursor in solution processing, we believe the novelty is there. The deadline limit of the revision hinders further exploration of the synthesis mechanism using high-end techniques. However, the data presented here sufficiently justifies our proposed mechanism. We appreciate the meticulous and comprehensive review which helped in further improving the manuscript. The new experiments suggested have actually helped us further in figuring out the material properties. Unlike conventional sensor, the H2O2 sensing activity is of ‘inhibitive’ character which is originating (probably) through the changes in electrode surface. Thus, the upper range will be limited in these materials.
Some comments that could be useful for the authors are:
- In the introduction, the real interests of measuring the hydrogen peroxide and the range of interest should be longer discussed. The authors do not properly present the necessity to develop a sensor to detect hydrogen peroxide, and the real limitations of current standard methods.
A: We have modified the introduction and discussion as suggested.
- In the introduction, the authors describe the disadvantages of using enzymatic methods to detect hydrogen peroxide and the advantages of using non-enzymatic methods. But they do not describe the mechanism of the hydrogen peroxide detection (potentials, why are interesting the use of bismuth, etc.). Some of the readers could not be familiarized with the electrochemical mechanism involved in the hydrogen peroxide detection. Please, you should clarify and extend this part too.
A: We have modified the introduction as suggested.
- In the synthesis section, the authors do not clarify why they are using these chemicals, concentrations and solutions. It looks as a standard method or something from a previous work. Please, add a reference or explain it in the text.
A: We have modified the manuscript as suggested. We have followed a protocol established in an earlier work [1] from our group, which was already mentioned in the last paragraph of the introduction.
- Chitara, B.; Limbu, T.B.; Orlando, J.D.; Vinodgopal, K.; Yan, F. 2-D Bi2O2Se Nanosheets for Nonenzymatic Electrochemical Detection of H2O2. IEEE Sensors Letters 2020, 4, doi:10.1109/LSENS.2020.3012300.
- Why are the authors using a glassy carbon electrode? What are the dimensions of the electrode? Why are thy using a purge solution to perform the electrochemical analysis? It could limit the real applicability in real scenarios of the sensor. What could happen if you do not purge the solution? The result and the applicability of the sensor could be reduced?
A: Glassy carbon is a commonly used low cost working electrode. The area of the electrode is 0.07 cm2 (3 mm diameter). Our sample also shows some oxygen reduction activity (generation of H2O2 from O2) with an onset close to -0.2 V vs Ag/AgCl (Fig. S9, ESI). The reduction peak, probably from the metal oxide/selenide is observed around -0.7 V vs Ag/AgCl. To differentiate and minimize the influence of in-situ produced H2O2 we have purged the solution with an inert gas. This is to ensure the current signals originate only from the spiked H2O2 solutions.
We have now also carried out DPV measurements without any purging (as suggested) for BOSe-6 h, as representative sample. The sensor still works well without any purging, in fact with better sensitivity.
- Why are you using those interferents to examinate the performance of the sensor? In which field are them interesting and in which range of concentrations? Clarify.
A: In biological fluids like sweat and urine, various analytes are commonly present. For non-invasive analysis of H₂O₂, which indirectly estimates glucose, electrochemical measurements in the presence of these interferants offer a more realistic comparison to real samples. Interferants can sometimes interact with the sensor material, generating additional peaks in CV curves. While an ideal sensor would be highly selective for a single analyte, this is often not the case in practice. Therefore, it is essential to analyze the electrochemical response of a sensor material to multiple analytes. For other applications, trace level detection of H2O2 can also be useful:
- Biomedical applications: 0.1–10 µM (oxidative stress and enzymatic reactions)
- Water quality monitoring: <10 µM (residual H₂O₂)
- Food safety: <5-10 µM (residual H₂O₂ in packaging)
- Pharmaceutical testing: 1-10 µM (ROS detection)
- Industrial quality control: <10 µM (residual H₂O₂ post-treatment)
We have included these details in discussion in the revised manuscript.
- How reproducible is the dispersion to drop cast the solution in the electrode? Is this homogenous enough to guarantee the sensor performance? Do you need to consider some tricks or hidden steps to do it?
A: The dispersion is quite homogenous and stable. In order to maintain reproducibility, we sonicate it for 15 min prior to each loading. Yes, we need to have some skill in drop casting within the working electrode area, as the dispersion spreads if its dilute. The use of combination of IPA/DI solution gives a good dispersion that doesn’t spread a lot. We also warm the electrode prior to drop casting the first drop, so it doesn’t spread out. These are some basic things which are usually followed in literature.
- In the Structural and Morphological Characterization section there are some sentences in italics. The authors should correct it.
A: We have corrected them as suggested.
- In the Figure 2, the scale and the conditions of the SEM analysis is too much small and impossible to read.
A: Separate scale bars are provided for clarity.
- Sensitivity and limit of detection are not necessary are independent equations in the manuscript. They are very common, so the authors should include it as part of the text.
A: We have incorporated the suggestion in the revised manuscript. However, upon further afterthought, we realize that the current response in DPV shows decreasing trend (inhibitive) with negative intercept. Since, the metal oxide/selenide itself has redox peaks for blank electrolyte (without H2O2) the limit of detection (LOD) isn’t meaningful here. So, we only describe the revised results in terms of sensitivity.
- For BOSe-6h, why the peak of the hydrogen peroxide is shifting for increasing concentrations? Clarify in the discussion.
A: The material undergoes redox reactions during each cycle. The CV scan starts from 0 V, and is scanned upto 0.8 V, followed by the scan in negative direction upto -0.9 V. The cycle is completed by scanning back to 0.8 V. We observe that for all samples, there is absence of the reduction peak in first CV cycle. Oxidation peaks appear in the first cycle itself. In the subsequent cycle, prominent reduction peak is observed. These reduction and oxidation peak gets intensified upon H2O2 additions. This suggests that some sort of chemical reactions occur between the electrolyte and electrode material. The redox peaks most probably belong to the metal oxide/selenide composite and isn’t necessarily from the H2O2 electroreduction. In presence of H2O2 the selenide component can get preferentially oxidized and lead to more intense peak for the oxide component. However, we are unable to conclusively prove this aspect at present will be studied in depth in future with high-end techniques. The shift in DPV peak position can also occur due to changes in the electrode material during each cycle. A few additional lines of discussion have been provided.
- In general, the discussion of the results is quite poor and short, as well as the conclusions. More discussions and explanations about the results should be incorporated to the manuscript.
A: The manuscript has been revised substantially as suggested. We have included more results and discussion to validate it.
Reviewer 3 Report
Comments and Suggestions for Authors
the authors of the manuscript nanomaterials-3185364 (“Electrochemical Detection of H2O2 using Bi2O3/Bi2O2Se Nano-composites”) reported modification of a previously published synthesis to prepare Bi2O3/Bi2O2Se nano-composites for the electrochemical detection of hydrogen peroxide. Unfortunately, I cannot recommend this manuscript for publication in its present form. The main problem is that the detection of hydrogen peroxide in this manuscript is performed at (–0.75) V, which is an unacceptably high absolute potential value for the sensor detection. There are many non-enzymatic sensors, which can detect hydrogen peroxide at potentials close to 0 V with high sensitivity, sufficient selectivity, and low detection limit. On the other hand, there is some sensitivity to the oxidation of hydrogen peroxide at about 0 V (Figure S1 d and e). Perhaps, the authors could try to evaluate the sensor properties of the Bi2O3/Bi2O2Se nano-composite based on the hydrogen peroxide oxidation near 0 V. Moreover, the influence of oxygen reduction might be less in this potential window. Possibly, the authors should focus on another application other than electrochemical detection of hydrogen peroxide. Further notes:
1) The authors focused on the electrochemical application of the composite for the H2O2 detection. Then, a better characterization of the sensor properties should be performed. In particular, the author should also study the response of the Bi2O3/Bi2O2Se modified electrode in the presence of oxygen.
2) Please, specify concentration of H2O2 in the selectivity studies, Figure 5. Figure 5 shows that other substances significantly influence the response, so the selectivity is low.
3) EDX: where did the large amount of C come from? Why is it not taken into account in the formula?
4) Figure order: Figure S1 (ESI) is mentioned in the text (section 3.3, p.6) after Figures S2 and S3 (section 3.1, p.4). Please, correct the order of the figures. Figures should be arranged as they are mentioned in the text of the manuscript.
Author Response
The authors of the manuscript nanomaterials-3185364 (“Electrochemical Detection of H2O2 using Bi2O3/Bi2O2Se Nano-composites”) reported modification of a previously published synthesis to prepare Bi2O3/Bi2O2Se nano-composites for the electrochemical detection of hydrogen peroxide. Unfortunately, I cannot recommend this manuscript for publication in its present form. The main problem is that the detection of hydrogen peroxide in this manuscript is performed at (–0.75) V, which is an unacceptably high absolute potential value for the sensor detection. There are many non-enzymatic sensors, which can detect hydrogen peroxide at potentials close to 0 V with high sensitivity, sufficient selectivity, and low detection limit. On the other hand, there is some sensitivity to the oxidation of hydrogen peroxide at about 0 V (Figure S1 d and e). Perhaps, the authors could try to evaluate the sensor properties of the Bi2O3/Bi2O2Se nano-composite based on the hydrogen peroxide oxidation near 0 V. Moreover, the influence of oxygen reduction might be less in this potential window. Possibly, the authors should focus on another application other than electrochemical detection of hydrogen peroxide.
We thank the reviewer for recommendation of our work and useful insights. We beg to differ with the reviewer on this. The mechanisms by which the H2O2 detection is carried out at 0 V (catalytic) differs from the sensing aspects discussed in this work. The interaction of H2O2 with these electrode materials modifies the structure itself which is observed as the current response variation. Although the sensing capabilities for several reported materials may be better, from the viewpoint of scientific curiosity, these Bi2O3/Bi2O2Se composites are interesting. There are no reports available of the H2O2 sensing for these materials. Thus, it’s interesting to observe the interactions of the analyte with the material.
We observe a very feeble DPV peak at 0 V, but the most significant changes occur at -0.7 V. Unlike CV, we only observe one clear peak, making the analysis simple. Furthermore, as observed in the test with different analytes, dopamine has a prominent DPV peak at 0 V, thus it would hinder estimation in practical or real samples. This signal is characteristic of the material which gets modified upon interaction with H2O2. Furthermore, the influence of ORR was eliminated by Ar-purging. The focus of this work is the alternative route and tunability in synthesis towards Bi2O2Se using Se powder, and the electrochemical changes that occur when interacted with H2O2. We can look into other possible applications in future
Further notes:
- The authors focused on the electrochemical application of the composite for the H2O2 Then, a better characterization of the sensor properties should be performed. In particular, the author should also study the response of the Bi2O3/Bi2O2Se modified electrode in the presence of oxygen.
A: We have carried out the electrochemical tests without purging and provided it in Fig. S8, ESI. We have also carried out ORR tests in presence of oxygen to get insights on the inherent H2O2 production with these samples.
- Please, specify concentration of H2O2 in the selectivity studies, Figure 5. Figure 5 shows that other substances significantly influence the response, so the selectivity is low.
A: The concentration of all analytes used was 10 mM stock (effective conc. 100 µM). In fresh experiments we have carried out tests with individual analytes too. These individual experiments confirm that compared to H2O2, the changes are less significant.
- EDX: where did the large amount of C come from? Why is it not taken into account in the formula?
A: The EDX and SEM analysis was done on Al-stub, with a carbon tape placed on it. On top of this, Ir-coated silica was used, where the powder samples were sprinkled. Therefore, the EDX analyses shows presence of these additional peaks. These are not from the material, as confirmed in XRD.
- Figure order: Figure S1 (ESI) is mentioned in the text (section 3.3, p.6) after Figures S2 and S3 (section 3.1, p.4). Please, correct the order of the figures. Figures should be arranged as they are mentioned in the text of the manuscript.
A: We apologize for this error and rectify these in revised manuscript.
Reviewer 4 Report
Comments and Suggestions for Authors
The paper entitled “Electrochemical Detection of H2O2 using Bi2O3/Bi2O2Se Nano-composites” want to prove that the high-performance hydrogen peroxide (H2O2) sensors are important subjects nowadays. Thus, various applications, including environmental monitoring, industrial processes, and biomedical diagnostics are focused on such electrical sensors. The present study wants to explores the development of H2O2 sensors based on bismuth oxide/bismuth oxyselenide (Bi2O3/Bi2O2Se) nanocomposites. The Bi2O3/Bi2O2Se nanocomposites were synthesized using a simple solution-processing method at room temperature, resulting in a unique heterostructure with possible electrocatalytic properties for H2O2 detection. Techniques such as powder X-ray diffraction (XRD), X-ray photoelectron spectroscopy (XPS) and scanning electron microscopy (SEM), was used for nanocomposites characterization. The synthesis time was varied to obtain different Se content of the nanocomposites.
Electrochemical measurements revealed that the Bi2O3/Bi2O2Se composite formed under different synthesis conditions displayed different sensitivity towards H2O2 detection, along with a wide linear detection range (0.02 - 15 μM) and low detection limit. The performance is attributed to the synergistic effect between Bi2O3 and Bi2O2Se, enhancing electron transfer and creating more active sites for H2O2 oxidation. These findings suggest that Bi2O3/Bi2O2Se nanocomposites have potential as H2O2 material sensors.
1. The novelty and advantages of the proposed work have to be clearly stated.
2. A table containing the electrode performances should be introduced. Moreover, a comparison with other performances of already published modified electrodes should be added and discussed (https://doi.org/10.3390/GELS10040230; https://doi.org/10.3390/gels9110868, https://doi.org/10.3390/molecules26010117; https://doi.org/10.1016/J.MATCHEMPHYS.2012.12.079
3. The authors, should not only present their results, they also should correlate them with other already published results (bibliography). (https://doi.org/10.3390/molecules26010117; https://doi.org/10.1016/J.MATCHEMPHYS.2012.12.079
4. During the electrochemical measurements, the authors used argon purging which was paused during the CV and DPV scans recording. Comparing with other already published works, which have no purging step, what are the performances (LOD, sensitivity, stability, reproducibility) of this work (https://doi.org/10.3390/molecules26010117; https://doi.org/10.1016/J.MATCHEMPHYS.2012.12.079, https://doi.org/10.3390/GELS10040230; https://doi.org/10.3390/gels9110868).
5. Another important parameter for H2O2 detection is the electroreduction potential of H2O2. Therefore, this parameter can be included in the summarised recommended Table.
6. A figure with the calibration curve and its linear regression it is necessary, for all modified electrodes. It is not clear whether the LOD and sensitivity was calculated from CVs or DPV.
7. The recorded CVs with and without H2O2 shows a reduction peak (around -0.8 V vs Ag/AgCl,KClsat) for all modified electrodes (excepted BOSe-18h). This means that it is something from the buffer solution or electrofde surface which is reduced at this potential.
8. In CV for BOSe 18h in the reduction part are two peaks (before and after -0.6 V). What means this peaks? Why these increase with the H2O2 concentration increasing?
9. The reproducibility, stability, and reusability of the obtained modified electrodes.
10. “The change in the redox peak patterns is clearly visible among the samples and also with increasing H2O2 additions.” The authors should explain all the peaks presence in the absence and presence of H2O2, and to propose an electroreduction mechanism for H2O2 at the electrode surface.
11. The interfering test should be corrected. There is no information on the H2O2 concentration. Moreover, the peak from -0.8 V seems to be changed in the presence of different interferes (UA, NaCl, ..). This is exactly the opposite effect – these prove that there are interferences.
12. The authors should read carefully the paper. Phrases like this: “Figure S1. CV scans of Bi2O2Se with synthesis time of (a) 10 min (b) 3 hours (c) 6 hours (d) 18 hours (e) 3 days and (f) 7 days.” should be corrected. There is no 3 days measurements.
13. In figure 3 (Tauc plots for (b) BOSe-6 h and (c) BOSe-7 days) y axes seems to be not correct calculated. The authors should check and correct the y values.
14. The authors affirm that the linear detection range is 0.02 - 15 μM. The calibration curve should confirms it.
15. All the results should be presented with errors and error bars (the presented results should be the mean of at least three measurements).
16. Why the authors choose the best electrode at 6h and not at 18h? The LOD is more than 2 times lower at 18h and the sensitivity is with 8% lower.
17. Real sample analysis should be included.
Comments on the Quality of English LanguageMinor editing of English language required.
Author Response
The paper entitled “Electrochemical Detection of H2O2 using Bi2O3/Bi2O2Se Nano-composites” want to prove that the high-performance hydrogen peroxide (H2O2) sensors are important subjects nowadays. Thus, various applications, including environmental monitoring, industrial processes, and biomedical diagnostics are focused on such electrical sensors. The present study wants to explores the development of H2O2 sensors based on bismuth oxide/bismuth oxyselenide (Bi2O3/Bi2O2Se) nanocomposites. The Bi2O3/Bi2O2Se nanocomposites were synthesized using a simple solution-processing method at room temperature, resulting in a unique heterostructure with possible electrocatalytic properties for H2O2 detection. Techniques such as powder X-ray diffraction (XRD), X-ray photoelectron spectroscopy (XPS) and scanning electron microscopy (SEM), was used for nanocomposites characterization. The synthesis time was varied to obtain different Se content of the nanocomposites.
Electrochemical measurements revealed that the Bi2O3/Bi2O2Se composite formed under different synthesis conditions displayed different sensitivity towards H2O2 detection, along with a wide linear detection range (0.02 - 15 μM) and low detection limit. The performance is attributed to the synergistic effect between Bi2O3 and Bi2O2Se, enhancing electron transfer and creating more active sites for H2O2 oxidation. These findings suggest that Bi2O3/Bi2O2Se nanocomposites have potential as H2O2 material sensors.
We thank the reviewer for recommendation of our work and useful insights.
- The novelty and advantages of the proposed work have to be clearly stated.
A: We have modified the manuscript as suggested.
- A table containing the electrode performances should be introduced. Moreover, a comparison with other performances of already published modified electrodes should be added and discussed (https://doi.org/10.3390/GELS10040230; https://doi.org/10.3390/gels9110868, https://doi.org/10.3390/molecules26010117; https://doi.org/10.1016/J.MATCHEMPHYS.2012.12.079
A: Table 1 includes the revised performance parameters of the electrodes. We include Table 2, showing comparison with published literature. The references have been cited and discussed.
- The authors, should not only present their results, they also should correlate them with other already published results (bibliography). (https://doi.org/10.3390/molecules26010117; https://doi.org/10.1016/J.MATCHEMPHYS.2012.12.079
A: The references have been cited and discussed.
- During the electrochemical measurements, the authors used argon purging which was paused during the CV and DPV scans recording. Comparing with other already published works, which have no purging step, what are the performances (LOD, sensitivity, stability, reproducibility) of this work.
(https://doi.org/10.3390/molecules26010117; https://doi.org/10.1016/J.MATCHEMPHYS.2012.12.079, https://doi.org/10.3390/GELS10040230; https://doi.org/10.3390/gels9110868).
A: The references have been cited and discussed. The purging step was carried out to ensure that there isn’t any influence from the intrinsic ORR activity. We have included the results without the purging step (Fig. S9, ESI) and the performance has been compared with these references.
- Another important parameter for H2O2 detection is the electroreduction potential of H2O2. Therefore, this parameter can be included in the summarised recommended Table.
A: A discussion on the electroreduction potential of H2O2 is provided in introduction. However, we are unsure about its clarity in our work and hence exclude this in Table 2.
- A figure with the calibration curve and its linear regression it is necessary, for all modified electrodes. It is not clear whether the LOD and sensitivity was calculated from CVs or DPV.
A: The calibration curves to evaluate LOD and sensitivity are analyzed for DPVs. It was already mentioned in text. The calibration curve for BOSe-6 h is now provided in Fig. 3(f). The calibration curves for all the other samples are provide in ESI (Fig. S6, ESI)
- The recorded CVs with and without H2O2 shows a reduction peak (around -0.8 V vs Ag/AgCl,KClsat) for all modified electrodes (excepted BOSe-18h). This means that it is something from the buffer solution or electrode surface which is reduced at this potential.
A: The reviewer has correctly pointed this out. We ourselves have been intrigued with this observation. We hypothesize that these redox peaks are from the metal oxide and metal chalcogenide. We observe that for all samples, there is absence of the reduction peak in first CV cycle. Oxidation peaks appear in the first cycle itself. In the subsequent cycle, prominent reduction peak is observed. These reduction and oxidation peak gets intensified upon H2O2 additions. This suggests that some sort of chemical reactions occur between the electrolyte and electrode material. The redox peaks most probably belong to the metal oxide/selenide composite and isn’t necessarily from the H2O2 electroreduction. In presence of H2O2 the selenide component can get preferentially oxidized and lead to more intense peak for the oxide component. However, we are unable to conclusively prove this aspect at present will be studied in depth in future with high-end techniques.
- In CV for BOSe 18h in the reduction part are two peaks (before and after -0.6 V). What means this peaks? Why these increase with the H2O2 concentration increasing?
A: We are still trying to figure this out. Our hypothesis is that the redox peaks at -0.64 and -0.54 V, belong to Bi2O3 and Bi2O2Se and get enhanced due to their interaction with H2O2. In presence of H2O2 the selenide component can get preferentially oxidized and lead to more intense peak for the oxide component.
- The reproducibility, stability, and reusability of the obtained modified electrodes.
A: We have now included the data for reproducibility (Fig. S7, Fig. 3f), and reusability (Fig. 4). The electrodes and dispersion are quite stable (we have them stored for more than 6 months) and the results are reproducible.
- “The change in the redox peak patterns is clearly visible among the samples and also with increasing H2O2” The authors should explain all the peaks presence in the absence and presence of H2O2, and to propose an electroreduction mechanism for H2O2 at the electrode surface.
A: We have included a mechanism in the manuscript. The difficulty arises with the composite structure, wherein the redox peaks for both oxide and chalcogenide appear at different potentials.
- The interfering test should be corrected. There is no information on the H2O2 Moreover, the peak from -0.8 V seems to be changed in the presence of different interferes (UA, NaCl, ..). This is exactly the opposite effect – these prove that there are interferences.
A: We have redone the interference tests with individual analytes. Each test was carried out with fresh electrodes and electrolyte solutions. The electrode actually change over multiple cycles, i.e., the material gets oxidized/reduced during the electrochemical cycling. Yes, there are some minor changes that occur with addition of various analytes. The absence of any significant change in current signals in the consecutive DPV tests indicates minimal role of interferants. Significant change is observed only in presence of dopamine and H2O2. Under normal concentrations usually biological samples (e.g. 2 µM) there is hardly any change. We have tried to use high conc. just to differentiate that there is “some’ interference from dopamine (in opposite way) which may not be observable under biological relevant concentration.
The concentration of the analytes for the earlier experiment was already mentioned as 10 mM stock.
- The authors should read carefully the paper. Phrases like this: “Figure S1. CV scans of Bi2O2Se with synthesis time of (a) 10 min (b) 3 hours (c) 6 hours (d) 18 hours (e) 3 days and (f) 7 days.” should be corrected. There is no 3 days measurements.
A: We apologize for this error. We have corrected this.
- In figure 3 (Tauc plots for (b) BOSe-6 h and (c) BOSe-7 days) y axes seems to be not correct calculated. The authors should check and correct the y values.
A: The Tauc plaot was indicative for ‘direct band gap’. We also include the Tauc plot for indirect ‘band gap’. The values were correctly mentioned as per the Kubelka-Munk formalism.
- The authors affirm that the linear detection range is 0.02 - 15 μM. The calibration curve should confirms it.
A: The calibration curves to evaluate sensitivity are analyzed for DPVs. It was already mentioned in text. The calibration curve for BOSe-6 h is now provided in Fig. 3(f). The calibration curves for all the other samples are provide in ESI (Fig. S6).
- All the results should be presented with errors and error bars (the presented results should be the mean of at least three measurements).
A: As the experiments take very long time, and constrained by the deadline of revision, we regret that multiple experiments for all samples could not be done. We include the reproducibility data and evaluate the mean results for BOSe-6 h as a representative study (Fig. S7, ESI).
- Why the authors choose the best electrode at 6h and not at 18h? The LOD is more than 2 times lower at 18h and the sensitivity is with 8% lower.
A: The DPV results clearly suggest that the most uniform increase in current signal occurs in BOSe-6 h sample. The results in Table 1 have been revised carefully to indicate linearity in the range 0.2 -15 µM. The sensitivity is optimum for 6 h sample. However, upon further afterthought, we realize that the current response in DPV shows decreasing trend (inhibitive) with negative intercept. Since, the metal oxide/selenide itself has redox peaks for blank electrolyte (without H2O2) the limit of detection (LOD) isn’t meaningful here. So, we only describe the revised results in terms of sensitivity. Nevertheless, the interference test carried thereafter is for a representative sample. We would still arrive at similar conclusion with other samples.
- Real sample analysis should be included.
A: We had already included the results in real urine sample for 2 individuals (Fig. S10).
Round 2
Reviewer 2 Report
Comments and Suggestions for Authors
The authors have addressed all my comments and suggestions. I would recommend the publication of the manuscript in the journal
Reviewer 3 Report
Comments and Suggestions for Authors
the authors have revised the manuscript. Although, in my opinion, the sensor application remains questionable, the manucript might be acceptable form other points of view.
Reviewer 4 Report
Comments and Suggestions for Authors
The paper entitled “Electrochemical Detection of H2O2 using Bi2O3/Bi2O2Se Nano-composites” want to develop high-performance hydrogen peroxide (H2O2) sensors which are important subjects nowadays. Various applications, such as environmental monitoring, industrial processes, and biomedical diagnostics are focused on using fast and competitive electrochemical sensors.
The present study developed modified electrodes based on bismuth oxide/bismuth oxyselenide (Bi2O3/Bi2O2Se) nanocomposites. The Bi2O3/Bi2O2Se nanocomposites were synthesized using a simple method at room temperature, resulting in a unique heterostructure with possible electrocatalytic properties for H2O2 detection.
Techniques such as powder X-ray diffraction (XRD), X-ray photoelectron spectroscopy (XPS) and scanning electron microscopy (SEM), was used for nanocomposites morpho-structural characterization. The synthesis time was varied in order to have different Se content of the nanocomposites.
The recorded electrochemical measurements revealed that the Bi2O3/Bi2O2Se composite formed under different synthesis conditions displayed different sensitivity towards H2O2 detection, along with a wide linear detection range (0.02 - 15 μM) and low detection limit. The performance is attributed to the synergistic effect between Bi2O3 and Bi2O2Se, enhancing electron transfer and creating more active sites for H2O2.
These findings suggest that Bi2O3/Bi2O2Se nanocomposites have potential as H2O2 material sensor.
Comments on the Quality of English LanguageMinor editing of English language